# Robust Multimodal Learning with Missing Modalities via Parameter-Efficient Adaptation

## Abstract

Multimodal learning seeks to utilize data from multiple sources to improve the overall performance of downstream tasks. It is desirable for redundancies in the data to make multimodal systems robust to missing or corrupted observations in some correlated modalities. However, we observe that the performance of several existing multimodal networks significantly deteriorates if one or multiple modalities are absent at test time. To enable robustness to missing modalities, we propose simple and parameter-efficient adaptation procedures for pretrained multimodal networks. In particular, we exploit low-rank adaptation and modulation of intermediate features to compensate for the missing modalities. We demonstrate that such adaptation can partially bridge performance drop due to missing modalities and outperform independent, dedicated networks trained for the available modality combinations in some cases. The proposed adaptation requires extremely small number of parameters (e.g., fewer than 0.7% of the total parameters in most experiments). We conduct a series of experiments to highlight the robustness of our proposed method using diverse datasets for RGB-thermal and RGB-Depth semantic segmentation, multimodal material segmentation, and multimodal sentiment analysis tasks. Our proposed method demonstrates versatility across various tasks and datasets, and outperforms existing methods for robust multimodal learning with missing modalities.

## 1 Introduction

Multimodal learning (MML) (Baltrušaitis et al. (2018); Xu et al. (2023)) is a general framework for processing, combining, and understanding information from multiple, diverse data sources. Fusing knowledge from multiple modalities (e.g., text, images, audio, and sensor data) is expected to provide more accurate and reliable systems. In recent years, MML has achieved remarkable success in a wide range of applications, including image segmentation (Chen et al. (2020); Wang et al. (2022); Zhang et al. (2023)), captioning (Zhao et al. (2019); Yu et al. (2019)), classification (Guillaumin et al. (2010); Roy et al. (2023)), sentiment analysis (Soleymani et al. (2017); Kaur & Kautish (2022)), and autonomous driving (Xiao et al. (2020); Rizzoli et al. (2022)). In all these applications, one often encounters situations where some modalities are corrupted or missing due to hardware limitations/failures or data acquisition cost/constraints. The ability to handle corrupt or missing modalities is thus crucial for the robustness and reliability of multimodal systems. The central question of this paper is to study and build robustness in MML with missing modalities.

Recent studies (Ma et al. (2022); Hazarika et al. (2022); McKinzie et al. (2023)) have shown that MML is not inherently robust to missing modalities and performance can drop significantly when modalities are missing at test time. Existing approaches for robust MML usually work for specific combinations of modalities they are trained for and tend to perform poorly when applied to untrained combinations. For instance, one approach is to adopt robust training strategies such as modality dropout during training (Neverova et al. (2015); Hussen Abdelaziz et al. (2020)), partial or full modality masking (Bachmann et al. (2022); Shin et al. (2023)), and knowledge distillation (Tarvainen & Valpola (2017); Maheshwari et al. (2023)). These approaches either require specialized training strategies or utilize extra models/sub-networks to guide the underlying model. Another approach replaces uninformative tokens with aggregated informative tokens from different modalities or learns to predict tokens for the specific missing modalities (Wang et al. (2022); Woo et al. (2023); Shin et al. (2023)). Training such separate (independent) networks for every possible modality combination is

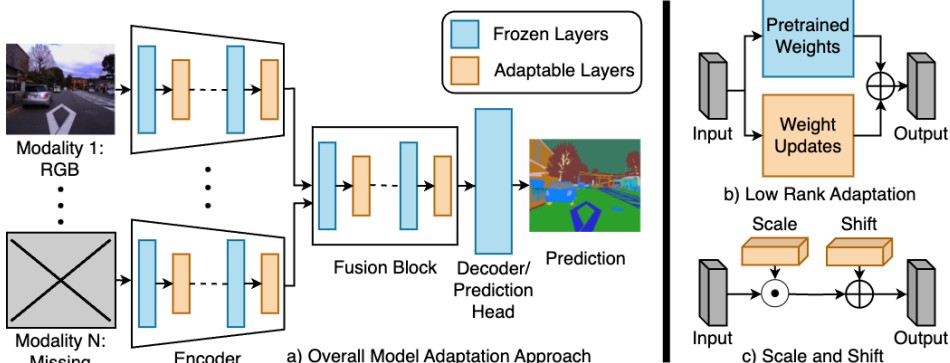

**Figure 1:** a) Overview of our approach for robust multimodal learning with missing modalities via parameter-efficient adaptation. A model pretrained on all the modalities is adapted using a small number of parameters to handle different modality combinations. b) Low-rank model adaption computes features using frozen and low-rank weights and combine them. c) Scale and shift feature adaptation transforms input by element-wise multiplication and addition. One set of parameters is learned for each modality combination.

not feasible. One recent approach for robust MML is to impute missing modalities from the available modalities (Yu et al. (2018); Sharma & Hamarneh (2019); Dorent et al. (2019)). Performance of these methods depend on the generation model that imputes the missing modalities.

In this paper, we propose a parameter-efficient approach to build a multimodal network that can adapt to arbitrary combinations of input modalities. Our main objective is to modify the network in a controllable manner as a function of available modalities. Figure 1 illustrates our proposed method, where a given multimodal network can be adapted to arbitrary modality combinations by transforming the intermediate features at different layers. To achieve parameter-efficient adaptation, we propose to use simple linear transformations such as scaling, shifting, or low-rank increments of features. Our method does not require retraining the entire model or any specialized training strategy. The adapted networks provide significant performance improvement over the multimodal networks trained with all modalities and tested with missing modalities. Performance of the adapted models is also comparable or better than the models that are exclusively trained for each input modality combination. We present a series of experiments to evaluate our method and compare with existing methods for robust MML. We tested different parameter-efficient adaptation strategies and found scaling and shifting features provides overall best performance with less than 0.7% of the total parameters.

**Contributions.** The main contributions can be summarized as follows.

- We propose parameter-efficient adaptation for multimodal learning that is robust to missing modalities. The adapted model can easily switch to different network states based on the available modalities with minimal latency, computational, or memory overhead.
- The adapted networks provide notably improved performance with missing modalities when compared to models trained with all modalities and is comparable to or better than the networks trained for specific modality combinations.
- Our approach is versatile and adaptable to a wide range of multimodal tasks and models. Detailed evaluations on diverse datasets and tasks show that our approach outperforms existing baseline methods and robust models designed for specific tasks and datasets.

## 2 RELATED WORK

**Multimodal learning with missing modalities** has been studied for different applications in recent years. For instance, robustness in vision-language tasks with multimodal transformers in Ma et al. (2022), multimodal sentiment analysis in McKinzie et al. (2023), multimodal classification in Hazarika et al. (2022), and multimodal action recognition in Woo et al. (2023). These studies have shown that the task performance can drop significantly when modalities are missing during test time.

**Robust training strategies** have been proposed to make models robust to different missing modalities. Such approaches include modality dropout during training Neverova et al. (2015); Hussen Abdelaziz

et al. (2020), unified representation learning Lau et al. (2019), and supervised contrastive learning Gomaa et al. (2022). Modality masking during training has become a popular choice for enhancing robustness. Shin et al. (2023) utilized complementary random masking, He et al. (2022) used masked auto encoder, and Ma et al. (2022) applied masked cross-modal attention for enhancing robustness of the underlying model. Hazarika et al. (2022) proposed noisy perturbation of modalities during training for robust multimodal sentiment analysis. Recently, Li et al. (2023) proposed uni-modal ensemble with modality drop and substitution augmentation during training to adapt to different missing modality scenarios.

**Design of robust models and fusion strategies** is another approach for robust MML. Fan et al. (2023) proposed a recursive meshing technique called SpiderMesh and Shin et al. (2023) designed complementary random masking and knowledge distillation based framework for robust RGB-thermal semantic segmentation. Wang et al. (2022) proposed TokenFusion to dynamically detect and replace uninformative tokens with projected tokens from other modalities for robust RGB-depth semantic segmentation, image-to-image translation, and 3D object detection. Wang et al. (2023) proposed a model that learns modality-shared and modality-specific features for robust brain tumour segmentation. Choi & Lee (2019) proposed a robust fusion strategy for multimodal classification. The main limitation of these methods is that they are generally designed for a specific modality combination and do not perform well when applied to other multimodal tasks Lin et al. (2023).

**Knowledge distillation and generation methods** have also become popular for robust MML. Studies by Sharma & Hamarneh (2019) and Yu et al. (2018) used GAN based generative models while Dorent et al. (2019) used VAE based generative models for imputing missing modalities from available input modalities for unlerlying multimodal tasks. Different knowledge distillation approaches have also been applied in several multimodal tasks. Tarvainen & Valpola (2017) proposed mean teacher and Maheshwari et al. (2023) introduced multimodal teacher for semi-supervised image segmentation. Shin et al. (2023) and Hazarika et al. (2022) applied self-distillation loss for robust RGB-thermal semantic segmentation. Apart from these approaches, weight space ensembling Wortsman et al. (2022), policy learning Ma et al. (2022) and optimal fusion strategy designing Maheshwari et al. (2023) were also studied for robust MML for various tasks.

## 3 PROPOSED METHOD

### 3.1 ADAPTATION FOR MISSING MODALITIES

Let us denote the set of input modalities for a given task as $\mathcal{M} = \{m_1, \ldots, m_M\}$. Given the full set $\mathcal{M}$, one can train a model $f$ with parameters $\Theta_{\mathcal{M}}$ that maps inputs for all the modalities (denoted as $\mathcal{X}_{\mathcal{M}}$) to an output $y_{\mathcal{M}}$ as

$$y_{\mathcal{M}} = f(\mathcal{X}_{\mathcal{M}}; \Theta_{\mathcal{M}}). \tag{1}$$

If a subset of the modalities $\mathcal{M}$ is missing, a naïve approach is to train a new model for the available input modalities. Without loss of generality, suppose $\mathcal{K} \subset \mathcal{M}$ represents missing modalities. We can use $\mathcal{S} = \mathcal{M} \setminus \mathcal{K}$ modalities to retrain the model $f$ for a new set of parameters $\Theta_{\mathcal{S}}$ as

$$y_{\mathcal{S}} = f(\mathcal{X}_{\mathcal{S}}; \Theta_{\mathcal{S}}), \tag{2}$$

where $\mathcal{X}_{\mathcal{S}}$ represents input data for modalities in $\mathcal{S}$. While we would prefer that $y_{\mathcal{S}} \approx y_{\mathcal{M}}$, but the training above does not guarantee it. In principle, we can train one model for every possible $\mathcal{S} \subset \mathcal{M}$ and use the corresponding model at the test time. Such an approach is infeasible because of computational and storage resources required to train models for a large number of possible modality combinations. Furthermore, deploying a large number of trained models and selecting one of them at test time is not feasible in real-world scenarios.

We propose an alternative approach to adapt a single model for all $\mathcal{S} \subset \mathcal{M}$ in a parameter-efficient manner. In particular, we train the model $f$ on the full set of modalities $\mathcal{M}$ and freeze the parameters $\Theta_{\mathcal{M}}$. We add a small number of parameters $\Delta_{\mathcal{S}}$, corresponding to the available modality set $\mathcal{S}$, and update the model as

$$\hat{y}_{\mathcal{S}} = f(\mathcal{X}_{\mathcal{S}}; \Theta_{\mathcal{M}}, \Delta_{\mathcal{S}}) \tag{3}$$

such that $\hat{y}_{\mathcal{S}} \approx y_{\mathcal{S}}$ in the worst case and $\hat{y}_{\mathcal{S}} \approx y_{\mathcal{M}}$ in the best case. Such an adaptation is parameter-efficient if the number of parameters in $\Delta_{\mathcal{S}}$ is significantly smaller than that in $\Theta_{\mathcal{M}}$. In our experiments, we keep $\Theta_{\mathcal{M}}$ frozen and demonstrate that less than $0.7\%$ of the total parameters for $\Delta_{\mathcal{S}}$ are sufficient for network adaptation.

## 3.2 METHODS FOR ADAPTATION

In recent years, a number of parameter-efficient methods have been proposed for network adaptation for various tasks. To the best of our knowledge, none of them have been applied for multimodal adaptation to handle missing modalities. In this section, we discuss some of the methods that we used to build our proposed adaptation method for robust MML with missing modalities.

**Low-rank adaptation (LoRA)** is one possible way to adapt models to missing modality scenarios. Such adaptation has been applied in other domains and tasks. For instance, LoRA in transformers Hu et al. (2022) and KAdaptataion He et al. (2023) learn low-rank factors for task/domain adaptation. In our context, suppose $W$ represents one of the weight matrices in the pretrained $\Theta_{\mathcal{M}}$. For a given $\mathcal{S}$, we can learn a low-rank matrix $W_{\mathcal{S}}$ for $\Delta_{\mathcal{S}}$. Since the update matrix is low-rank, the number of parameters needed for $\Delta_{\mathcal{S}}$ remains a fraction of that in $\Theta_{\mathcal{S}}$.

**Scaling and shifting features (SSF)** is another parameter-efficient method to transform intermediate features of the pretrained model Ioffe & Szegedy (2015); Lian et al. (2022). As shown in Figure 1c, SSF applies a linear transformation to the given token/feature with learnable scale ($\gamma$) and shift parameters ($\beta$). Given an input token $x$, SSF generates the output token $h$ as $h = \gamma \odot x + \beta$, where $\gamma, \beta, x, h$ are vectors of same length and $\odot$ represents element-wise multiplication along the embedding dimension. These scale and shift parameters are input-independent, meaning they are applied to the features regardless of the specific input modality. They are learned during the training process to help the model adjust and fine-tune its representations for better performance on the underlying task.

**BitFit** in Ben Zaken et al. (2022) and **Norm** adaptation (both batch norm and layer norm) are other approaches that adapt a subset of the model parameters. BitFit adapts the bias terms and Norm adaptation adapts norm layers, while keeping everything else frozen. Our experiments show that intermediate feature modulation via these simple linear transformation works well for most of the scenarios.

## 3.3 OUR APPROACH: PRAMETER-EFFICIENT ADAPTATION FOR MISSING MODALITIES

Our overall approach for model adaptation is illustrated in Figure 1. We first train a network with all available modalities in $\mathcal{M}$ and freeze the weights $\Theta_{\mathcal{M}}$. To adapt the model for different $\mathcal{S} \subset \mathcal{M}$, we insert SSF layers after each linear, convolutional, and norm (both batch norm and layer norm) layers. We learn $(\gamma, \beta)$ for all the SSF layers for given $\mathcal{S}$. While training SSF parameters for the given modality combination, $\mathcal{S}$, we set the missing modalities to zero. At the test time, we can easily change the SSF parameters corresponding to the available modalities. We only insert SSF layers in the encoder and fusion blocks, while keeping the decoder/prediction head unchanged. We observed that using pretrained decoder/prediction head provided a good overall performance with several missing modalities.

We primarily selected the SSF technique for robust multimodal learning with missing modalities because of its simplicity and effectiveness. SSF was introduced in Ioffe & Szegedy (2015) with batch normalization to potentially enhance the representation power of the networks and faster convergence. Ba et al. (2016) used the same strategy for layer normalization and Lian et al. (2022) used it for fine-tuning pretrained models for different image classification tasks on several datasets. SSF offers several benefits: First, the parameters $(\gamma, \beta)$ are independent of the input features, which makes SSF applicable to diverse tasks and input modality combinations. Second, we can easily insert SSF layers in the existing model without changing the model architecture. We can easily switch/select the corresponding SSF parameters for a given input modality combination. Finally, SSF introduces extremely small number of additional learnable parameters. The resulting adaptation offers significant savings compared to training a separate model for each input combination or retraining the model using some specialized training strategy like modality dropout or knowledge distillation.

## 4 EXPERIMENTS AND RESULTS

We performed detailed experiments to evaluate the performance of our proposed method for different tasks and datasets. We also present comparison with existing methods that are robust to missing modalities.

## 4.1 DATASETS

**Multimodal segmentation.** We used three datasets for three multimodal segmentation tasks. MFNet with RGB-Thermal images Ha et al. (2017), NYUDv2 with RGB-Depth images Silberman et al. (2012) and MCubeS for multimodal material segmentation with RGB, Angle of Linear Polarization (AoLP), Degree of Linear Polarization (DoLP) and Near-Infrared (NIR) images Liang et al. (2022). These datasets are divided into train and test sets along with ground truth per-pixel annotation for the underlying segmentation tasks.

**Multimodal sentiment analysis.** CMU-MOSI from Zadeh et al. (2016) and CMU-MOSEI from Bagher Zadeh et al. (2018) are two popular datasets for multimodal sentiment analysis using audio, visual and text as input modalities. They contain 2199 and 23453 annotated data samples, respectively, divided into train, validation, and test sets.

## 4.2 IMPLEMENTATION DETAILS

To investigate missing modality adaptation performance in multimodal semantic and material segmentation tasks, we use the CMNeXt model by Zhang et al. (2023) as the base model. We use multimodal transformer by Tsai et al. (2019) as the base model for multimodal sentiment analysis. We train a base model with all the modalities for each dataset. To evaluate performance with missing modalities, we provide the available modalities and set the missing modalities to zero. To perform model adaptation for any modality subset $\mathcal{S} \subset \mathcal{M}$, we freeze the pretrained weights and insert learnable SSF layers. Then we fine tune the learnable parameters for 100 epochs for multimodal segmentation tasks and until convergence for multimodal sentiment analysis tasks.

For multimodal segmentation tasks, we set the initial learning rate to $6 \times 10^{-5}$ and applied a polynomial learning rate scheduler with a power of 0.9. The first 10 epochs were set as the warm-up, during which the learning rate was set to 0.1 times the original rate. The scale parameters ($\gamma$) were initialized with all 1s and the shift parameters ($\beta$) were initialized with all 0s. We used cross-entropy loss function and AdamW optimizer as proposed in Loshchilov & Hutter (2019), with an epsilon value of $10^{-8}$ and a weight decay of 0.01. We used a batch size of 4 and report single scale performance for all the datasets. All other hyper-parameters and configurations are the same as Zhang et al. (2023). For multimodal sentiment analysis tasks, we used the default settings for the datasets as configured in the codebase Yu et al. (2021). We have included additional details for each dataset and experimental setup in the supplementary section.

## 4.3 BASELINE METHODS

We report experiments and results for different methods that are listed as follows. **Pretrained** model refers to the base model that is trained with all the available modalities. **Dedicated** training refers to independent models trained for each input modality combination. **Adapted** model refers to the model that is adapted using our approach for each input modality combination.

**Different robust methods** have been proposed for different multimodal tasks. We compare our method with the following methods: SpiderMesh Fan et al. (2023), VPFNet Lin et al. (2023), MDRNet Zhao et al. (2023), CRM Shin et al. (2023) for robust RGB-thermal semantic segmentation. CEN Wang et al. (2020a), TokenFusion Wang et al. (2022), AsymFusion Wang et al. (2020b), Dialated FCN-2s Kamran & Sabbir (2018) for robust RGB-depth semantic segmentation.

For every task/dataset, we adopted the experimental setup used in the corresponding previous studies. We used the reported results from prior works where possible. It is important to note that, because of this criteria, some of the baseline methods may only be present in specific experiments depending on the availability of their reported numbers. We also perform detailed comparison analysis of SSF with other parameter-efficient adaptation techniques. For all the experiments we follow the same setup suggested by the corresponding papers and report mean accuracy, F1 score and mean intersection over union (mIoU) when available.

**Ablation studies and comparison of different parameter-efficient adaptation methods** show that SSF-based adaptation provides overall best performance. We present results for scale only, shift only, BitFit (Ben Zaken et al. (2022)), norm layer fine-tuning and LoRA (Hu et al. (2022)).

**Table 1:** Performance of Pretrained, Dedicated, and Adapted networks with missing modalities. CMNeXt is the base model for multimodal semantic segmentation for MFNet and NYUDv2 datasets and multimodal material segmentation for MCubeS dataset. HHA-encoded images were used instead of raw depth maps. **Bold** letters represent best results.

| Dataset | Input | Missing | Pretrained | Dedicated | Adapted |
|---|---|---|---|---|---|
| MFNet | RGB-Thermal | - | 60.10 | 60.10 | - |
| | RGB | Thermal | 53.71 | **55.86** | 55.22 |
| | Thermal | RGB | 35.48 | **53.34** | 50.89 |
| NYUDv2 | RGB-Depth | - | 56.30 | 56.30 | - |
| | RGB | Depth | 51.19 | 52.18 | **52.82** |
| | Depth | RGB | 5.26 | 33.49 | **36.72** |
| MCubeS | RGB-AoLP-DoLP-NIR | - | 51.54 | 51.54 | - |
| | RGB-AoLP-DoLP | NIR | 49.06 | 49.48 | **51.11** |
| | RGB-AoLP | DoLP-NIR | 48.81 | 48.42 | **50.62** |
| | RGB | AoLP-DoLP-NIR | 42.32 | 48.16 | **50.43** |

## 4.4 EXPERIMENTS FOR MULTIMODAL SEGMENTATION

In this section, we present experiments for multimodal semantic segmentation with RGB-Thermal and RGB-Depth datasets, and multimodal material segmentation with MCubeS dataset. We report the detailed results for multimodal material segmentation along with per class % intersection over union (IoU) comparisons between the Pretrained and Adapted models in the supplementary section.

**Overall performance comparison.** We present the performance comparison of Pretrained, Dedicated, and Adapted networks for different missing modalities in Table 1. We observe that the performance of the Pretrained model drops significantly with missing modalities. We see a 6.39% drop when Thermal is missing in MFNet dataset and 5.11% drop when Depth is missing in NYUDv2 dataset compared to the case when all modalities are available. The effect is amplified when RGB gets missing as we observe 24.62% drop in MFNet dataset and 51.04% drop in NYUDv2 dataset. In MCubeS dataset, we observe 2.48–9.22% drop in pretrained model when different modality combinations are missing.

The overall performance of Adapted models with missing modalities is significantly better than Pretrained models. For MFNet, an improvement of 1.51% compared to the Pretrained model when RGB is available and thermal is missing. The Adapted model performance is close to the performance of Dedicated network trained for RGB only. The adapted model shows a significant improvement of 15.41% compared to the Pretrained model when RGB is missing. For NYUDv2 dataset, we see 1.63% and 31.46% performance improvement compared to Pretrained model when depth and RGB are missing, respectively. In both cases, the performance of the Adapted model is better than the Dedicated model. For all input combinations in MCubeS dataset, we see 1.82–8.11% performance improvement compared to the Pretrained model. The Adapted model performs better than Dedicated models in all the cases.

**Visualization of segmentation maps.** For qualitative analysis, we show some examples of the predicted segmentation maps form the Pretrained and Adapted models in Figure 2. For each dataset, we show the input images, predictions when all the modalities are available during both training and test time (CMNeXt column), predictions from the pretrained and adapted models for different available/missing modality scenarios (Available input modality names are shown in parentheses above each image). We see in Figure 2a, the Pretrained model fails to detect humans when only RGB is available and cars when only Thermal images are available. The adapted model can detect both humans and cars with missing modalities. For NYUDv2 dataset, as shown in Figure 2b, the Adapted model can detect window, bed, and furniture with higher accuracy than the Pretrained model with missing modalities. We only show the RGB input images for MCubeS dataset for brevity in Figure 2c. The Adapted model can identify sand, sky, and gravel with higher accuracy than the pretrained model. In all cases, the predictions from the Adapted model with missing modalities are

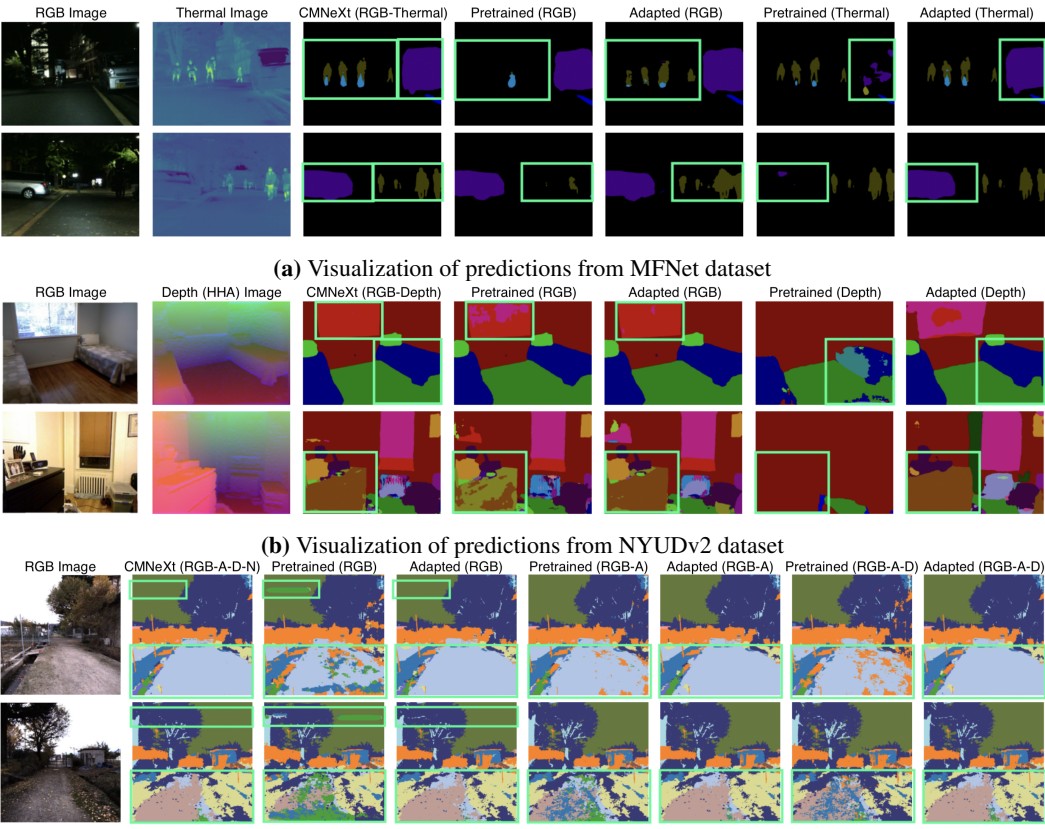

RGB Image | Thermal Image | CMNeXt (RGB-Thermal) | Pretrained (RGB) | Adapted (RGB) | Pretrained (Thermal) | Adapted (Thermal)

**(a)** Visualization of predictions from MFNet dataset

RGB Image | Depth (HHA) Image | CMNeXt (RGB-Depth) | Pretrained (RGB) | Adapted (RGB) | Pretrained (Depth) | Adapted (Depth)

**(b)** Visualization of predictions from NYUDv2 dataset

RGB Image | CMNeXt (RGB-A-D-N) | Pretrained (RGB) | Adapted (RGB) | Pretrained (RGB-A) | Adapted (RGB-A) | Pretrained (RGB-A-D) | Adapted (RGB-A-D)

**(c)** Visualization of predictions from MCubeS dataset

**Figure 2:** Examples of predicted segmentation maps for the Pretrained and Adapted models for multimodal semantic segmentation (on MFNet and NYUDv2) and material segmentation (on MCubeS). Title above each subimage shows method name (tested with available modalities). CMNeXt column shows the predictions with all the modalities. Segmentation quality improves significantly after model adaptation for all the input modality combinations. Green boxes highlight areas with salient differences in results (e.g., cars and humans missing in the Pretrained model with missing modalities but visible in the Adapted model). For MCubeS dataset, we only show RGB images and A, D and N denote angle of linear polarization, degree of linear polarization, and near-infrared, respectively.

closer to the predictions of the pretrained model with all modalities. We provide additional examples in the supplementary section.

**Performance Comparison for RGB-Thermal Semantic Segmentation.** We compare the performance of the Adapted model with existing robust models for RGB-thermal semantic segmentation with MFNet dataset in Table 2. The results show that the Adapted model offers the best average performance compared to the other baseline models, in terms of mean accuracy and % mIoU. Among the robust models, CRM Shin et al. (2023) shows competitive performance with the Adapted model. The Adapted model performs better when only RGB is available while CRM performs better when only Thermal is available. Notably CRM is designed specifically for RGB-Thermal pairs and requires specialized training approach and utilizes self-distillation loss between the clean and masked modalities to train the model. In contrast, our approach is applicable to any input modalities and does not require any specialized training technique.

**Performance Comparison for RGB-Depth Semantic Segmentation.** Table 3 shows the performance comparison with existing robust models for RGB-Depth semantic segmentation on NYUDv2 dataset. The table shows that on average, the Adapted model performs better than the existing robust models. TokenFusion Wang et al. (2022) performs slightly better (+0.12%) in terms of mIoU when Depth is available and RGB is missing, but shows larger drop (-5.59%) in mean accuracy. On the other hand, the Adapted model performs significantly better (+3.5% mIoU and +4.47% mean

**Table 2:** Performance comparison with existing robust methods for MFNet dataset. RGB and Thermal columns report performance when only RGB and Thermal are available. Average column reports average performance when one of the two modalities gets missing. '-' indicates that results for those cells are not published. Mean accuracy (mAcc) and % mean intersection over union (mIoU) are shown for all the experiments.

| Methods | RGB | | Thermal | | Average | |
|---|---|---|---|---|---|---|
| | mAcc | % mIoU | mAcc | % mIoU | mAcc | % mIoU |
| FuseNet (Hazirbas et al. (2016)) | 11.11 | 10.31 | 41.33 | 36.85 | 26.22 | 23.58 |
| MFNet (Ha et al. (2017)) | 26.62 | 24.78 | 19.65 | 16.64 | 23.14 | 20.71 |
| RTFNet (Sun et al. (2019)) | 44.89 | 37.30 | 26.41 | 24.57 | 35.65 | 30.94 |
| SAGate (Chen et al. (2020)) | 32.01 | 30.57 | 13.34 | 12.51 | 22.68 | 21.54 |
| FEANet (Deng et al. (2021)) | 15.96 | 8.69 | 58.35 | 48.72 | 37.16 | 28.71 |
| MDRNet (Zhao et al. (2023)) | 57.11 | 45.89 | 41.98 | 30.19 | 49.55 | 38.04 |
| VPFNet (Lin et al. (2023)) | 48.14 | 41.08 | 42.20 | 35.80 | 45.17 | 38.44 |
| SpiderMesh (Fan et al. (2023)) | - | 39.60 | - | 50.50 | - | 45.05 |
| CRM (Shin et al. (2023)) | - | 52.70 | - | **53.10** | - | 52.90 |
| Adapted (Ours) | **67.18** | **55.22** | **66.70** | 50.89 | **66.94** | **53.06** |

**Table 3:** Performance comparison with existing robust methods for NYUDv2 dataset. RGB and Depth columns report performance when only RGB and Depth are available. Average column indicates average performance when one of the two modalities gets missing. ∗ indicates that available codes and pretrained models from the authors were used to generate the results. Other results are from the corresponding papers.

| Methods | RGB | | Depth | | Average | |
|---|---|---|---|---|---|---|
| | mAcc | % mIoU | mAcc | % mIoU | mAcc | % mIoU |
| FCN (Long et al. (2015)) | 44.70 | 31.60 | 35.70 | 25.20 | 40.20 | 28.40 |
| Dilated FCN-2s (Kamran & Sabbir (2018)) | 47.10 | 32.30 | 39.30 | 26.80 | 43.20 | 29.55 |
| AsymFusion (R-101) (Wang et al. (2020b)) | 59.00 | 46.50 | 45.60 | 34.30 | 52.30 | 40.40 |
| CEN (R-101) (Wang et al. (2020a)) ∗ | 51.77 | 39.59 | 28.98 | 19.32 | 40.38 | 29.46 |
| TokenFusion (Wang et al. (2022)) ∗ | 63.49 | 49.32 | 46.83 | **36.84** | 55.16 | 43.08 |
| Adapted (Ours) | **67.96** | **52.82** | **52.42** | 36.72 | **60.19** | **44.77** |

accuracy) when RGB is avalable and Depth is missing. The average performance of the Adapted model is also better than the TokenFusion model despite the fact that TokenFusion was designed to work with RGB-Depth pair, whereas the Adapted method is independent of input modalities. For our experiments, we use HHA-encoded images proposed by Gupta et al. (2014) instead of raw depth maps.

**Comparison with Parameter Efficient Model Adaption Techniques.** Apart from robust models, we also compare different parameter-efficient adaptation techniques. We report the results in Table 4. For MFNet dataset, SSF outperforms all the methods and performance is significantly better than the Pretrained model and close to the Dedicated models. For NYUDv2 and MCubeS datasets, the Adapted model performs better than both Pretrained and Dedicated models. These experiments also show that SSF performs better than other methods for most of the input modality combinations for all the datasets. We show a detailed comparison for each dataset in terms of mean accuracy, F1 score and % mIoU in the supplementary section of this paper.

## 4.5 EXPERIMENTS FOR MULTIMODAL SENTIMENT ANALYSIS

We tested our adaptation method for a multimodal sentiment analysis task and report the results in Table 5. We used the multimodal transformer (MulT) in Tsai et al. (2019) as the base model, and the CMU-MOSI dataset in Zadeh et al. (2016) and the CMU-MOSEI dataset in Bagher Zadeh et al. (2018) for evaluation. A, V and T stand for audio, video and text modalities, respectively. We observed that when text is available and either audio or video or both are missing at the test time, the performance does not drop significantly. Similar trend was reported in Hazarika et al. (2022). If text is missing at test time, then the performance of the Pretrained models drops significantly. The Adapted models can partially compensate for missing text and offer significantly better performance.

**Table 4:** Performance comparison (% mIoU) of different parameter-efficient adaptation techniques for MFNet, NYUDv2, and MCubeS datasets. Each column reports mIoU of the Adapted model with the corresponding modalities, and Avg indicates average performance. A and D denote Angle and Degree of Linear Polarization.

| Datasets | MFNet | | | NYUDv2 | | | MCubeS | | | |
|---|---|---|---|---|---|---|---|---|---|---|
| Methods | RGB | Thermal | Avg | RGB | Depth | Avg | RGB | RGB-A | RGB-A-D | Avg |
| Pretrained | 53.71 | 35.48 | 44.60 | 51.19 | 5.26 | 28.23 | 42.32 | 48.81 | 49.06 | 46.73 |
| Dedicated | 55.86 | 53.34 | 54.60 | 52.18 | 33.49 | 42.84 | 48.16 | 48.42 | 49.48 | 48.69 |
| Scale Only | 54.77 | 49.23 | 52.00 | 53.04 | 36.12 | 44.58 | 50.16 | 50.55 | **51.13** | 50.61 |
| Shift Only | 54.57 | 48.96 | 51.77 | 53.04 | 36.25 | 44.65 | 50.13 | 50.40 | 50.86 | 50.46 |
| BitFit | 54.39 | 49.07 | 51.73 | **53.09** | 36.64 | **44.87** | 50.19 | 50.57 | 51.07 | 50.61 |
| LoRA | 54.19 | 47.45 | 50.82 | 52.87 | 34.97 | 43.92 | 49.59 | 50.07 | 50.80 | 50.15 |
| Norm | 54.65 | 47.49 | 51.07 | 53.05 | 34.73 | 43.49 | 49.95 | 50.51 | 51.07 | 50.51 |
| Scale and Shift | **55.22** | **50.89** | **53.06** | 52.82 | **36.72** | 44.77 | **50.43** | 50.62 | 51.11 | **50.72** |

**Table 5:** Performance of Pretrained and Adapted models for multimodal sentiment analysis with CMU-MOSI and CMU-MOSEI datasets. Multimodal Transformer (MulT) as the base model. A, V and T denote audio, video, and text, respectively. ↑ means higher is better and ↓ means lower is better. The Adapted model outperforms the Pretrained model with missing modalities.

| Dataset | Input | Missing | Pretrained | | | Adapted | | |
|---|---|---|---|---|---|---|---|---|
| | | | Acc ↑ | F1 ↑ | MAE ↓ | Acc ↑ | F1 ↑ | MAE ↓ |
| CMU-MOSI | AVT | - | 79.88 | 79.85 | 0.918 | - | - | - |
| | AV | T | 48.93 | 41.95 | 1.961 | **55.49** | **53.96** | **1.469** |
| | A | VT | 48.32 | 40.98 | 1.875 | **50.00** | **46.71** | **1.464** |
| | V | AT | 52.44 | 51.77 | 1.547 | **54.88** | **54.39** | **1.471** |
| CMU-MOSEI | AVT | - | 83.79 | 83.75 | 0.567 | - | - | - |
| | AV | T | 41.91 | 32.78 | 1.025 | **63.32** | **60.69** | **0.819** |
| | A | VT | 37.15 | 20.12 | 1.089 | **62.85** | **55.55** | **0.826** |
| | V | AT | 38.28 | 23.70 | 1.075 | **62.49** | **60.00** | **0.819** |

For CMU-MOSI dataset, we see a significant 30.95% drop in accuracy when text is missing for the pretrained model compared to the case when all modalities are available. The Adapted model offers 6.56% improvement in accuracy over the Pretrained model. We also observe 1.68% and 2.44% improvement in accuracy over the Pretrained model performance when only audio and only video are available, respectively. In all these scenarios, we also see larger improvement in F1 score and reduction in mean absolute error (MAE). For CMU-MOSEI dataset, we see even greater improvement in all the metrics. Experiments show 21.41%, 25.7% and 24.21% improvement in accuracy for audio-video, audio only, and video only scenarios compared to the Pretrained model. We also observe 27.91%-36.30% improvement in F1 score and 0.206-0.263 reduction in mean absolute error (MAE).

# 5 CONCLUSION

Missing modalities at test time can cause significant degradation in the performance of multimodal systems. In this paper, we presented a simple and parameter-efficient adaptation method for robust multimodal learning with missing modalities. We demonstrated that simple linear operations can efficiently transform a single pretrained multimodal network and achieve performance comparable to multiple (independent) dedicated networks trained for different modality combinations. We evaluated the performance of our method and compared with existing robust methods for different multimodal segmentation and sentiment analysis tasks. Our method requires an extremely small number of additional parameters (e.g., $< 0.7\%$ of the total parameters in most experiments), while significantly improving performance compared to missing modality scenarios. Our adaptation strategy is applicable to different network architectures and tasks, which can be a versatile solution to build robust multimodal systems.

## REPRODUCIBILITY STATEMENT

We are committed to ensuring the reproducibility of our research and to facilitate the broader scientific community in replicating and building upon our work. The source code and trained models are available at this anonymous link. We have provided a clear and comprehensive README.md file to guide users in setting up the environment, running the code, and reproducing the results in the paper. We outline the specific data preprocessing steps, list of hyperparameters and configurations used in our experiments in Section 4.2 in the main text and Section B in the supplementary section. We hope this makes it easy for others to replicate our experiments. We have provided scripts and instructions in our source code to reproduce the main experimental results presented in this paper. Additionally, we have provided pretrained models allowing others to directly reproduce the results.

## ETHICS STATEMENT

To the best of our knowledge this work does not give rise to any significant ethical concerns.

## ACKNOWLEDGEMENT

This work is supported in part by AFOSR award FA9550-21-1-0330.

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

# SUPPLEMENTARY MATERIAL

## A  DATASETS

**MFNet Dataset** introduced by Ha et al. (2017), is a popular dataset for RGB-thermal urban scene segmentation, particularly in the context of supporting autonomous driving applications. It comprises a total of 1569 aligned pairs of RGB-thermal images. Within this collection, 820 image pairs were captured during daytime, while 749 pairs were acquired during nighttime. The dataset is divided into distinct training and test sets, each accompanied by pixel-level annotations that define semantic labels for nine classes. Each image is $640 \times 480$ pixels.

**NYU Depth v2 (NYUDv2) Dataset** from Silberman et al. (2012) is a well-known dataset for RGB-D semantic segmentation. This dataset contains 1449 pairs of aligned RGB-depth images of indoor scenes. The images are divided into training and test sets containing 795 and 654 pairs of images respectively. The dataset also provides per pixel annotations for 13 classes, 40 classes and 894 classes ground truth semantic labels. For our experiments we used the standard 40 classes annotation. Each image is $640 \times 480$ pixels and the dataset contains both raw and processed depth maps. For our experiments we used HHA images as proposed by Gupta et al. (2014) instead of depth maps.

**Multimodal Material Segmentation (MCubeS) Dataset** was introduced by Liang et al. (2022) for accurate multimodal material segmentation with the help of thermal and polarized images alongside RGB images. This dataset has four modalities: RGB, Angle of Linear Polarization, Degree of Linear Polarization and Near-Infrared. Alongside these modalities, the dataset also provides ground truth annotation for semantic and material segmentation. There are 500 image sets divided into train, validation and test sets having 302, 96 and 102 image sets respectively. The images are $1224 \times 1024$ pixels each and have 20 classes in total.

**CMU-MOSI** dataset from Zadeh et al. (2016) is a popularly used for multimodal sentiment analysis. The dataset has 2199 samples each having audio, visual and text as input modalities. It is divided into train, validation and test sets containing 1284, 229 and 686 samples respectively along with annotated sentiment for each sample.

**CMU-MOSEI** is a large scale sentiment analysis dataset from Bagher Zadeh et al. (2018). It is 10 times larger than CMU-MOSI and contains audio, visual and text modalities along with ground truth sentiment annotations. The dataset contains 23453 samples divided into train, validation and test sets for multimodal sentiment analysis and emotion recognition.

## B  IMPLEMENTATION DETAILS

We used Python[1] 3.8.12 and PyTorch[2] 1.9.0 to for our implementation. The experiments were done using two NVIDIA RTX 2080 Ti GPUs. We applied automatic mixed precision (AMP) training provided by PyTorch. For CMNeXt model, we use their publicly available code[3] and models trained on all the available modalities for each dataset. We trained the multimodal transformer models on all the modalities using the available code and preprocessed data from the repository[4] for CMU-MOSI and CMU-MOSEI datasets.

**MFNet Dataset:** We divied the 4 channel RGB-T images into three channel RGB and one channel thermal images. Then data pre-processing and augmentation was applied following CMNeXt from Zhang et al. (2023). MiT-B4 from Xie et al. (2021) was the backbone for the base CMNeXt model. One set of scale and shift parameters was learnt for each input modality combination. Input images were sized at $640 \times 480$ for both training and testing and we report single scale performance for all the experiments. The scale and shift parameters were trained for 100 epochs with a batch size of 4.

**NYUDv2 Dataset:** For processing depth maps, we follow SA-Gate by Chen et al. (2020) and CMNeXt by Zhang et al. (2023) and use HHA-encoded images instead of raw depth maps. The

---

[1] https://www.python.org/
[2] https://pytorch.org/
[3] https://github.com/jamycheung/DELIVER
[4] https://github.com/thuiar/MMSA

**Table 6:** Hyperparameters for the experiments on CMU-MOSI and CMU-MOSEI datasets for multimodal sentiment analysis.

| Hyperparameters | CMU-MOSI | CMU-MOSEI |
|---|---|---|
| Batch Size | 16 | 4 |
| Initial Learning Rate | 0.002 | 0.0005 |
| Optimizer | Adam | Adam |
| Attention Dropout | 0.3 | 0.4 |
| Embedding Dropout | 0.2 | 0.0 |
| Output Dropout | 0.5 | 0.5 |
| Gradient Clip | 0.6 | 0.6 |
| Weight Decay | 0.005 | 0.001 |
| Temporal Conv Kernel Size (T/A/V) | 5/5/5 | 5/1/3 |
| # of Crossmodal Blocks | 4 | 4 |

**Table 7:** Learnable parameter counts for different parameter efficient model adaptation methods. As seen from the table, scale and shift introduce less than 0.7% of the total model parameters.

| Method | Total Parameters (M) | Learnable Parameters (M) | % of Total Parameters |
|---|---|---|---|
| Norm | 116.560 | 0.126 | 0.108 |
| BitFit | 116.560 | 0.378 | 0.324 |
| LoRA | 116.957 | 0.397 | 0.340 |
| Scale and Shift | 117.349 | 0.789 | 0.673 |

already preprocessed dataset can be downloaded from the SA-Gate repository[5]. RGB and HHA images were sized at $640 \times 480$ pixels each and we used this size for training and testing. The backbone was set to MiT-B4 as suggested in CMNeXt paper. One set of scale and shift parameters was learnt for each input modality combination by feeding available input modalities and setting the missing modality to zero. We train the scale and shift parameters for 100 epochs with a batch size of 4 and report single scale performance.

**MCubeS Dataset:** We follow the same data pre-processing and augmentations used by the base CMNeXt model from Zhang et al. (2023). MiT-B2 from Xie et al. (2021) was used as the backbone for this dataset. We set the input image resolution to $512 \times 512$ during training and $1024 \times 1024$ during testing and report single scale performance with predicted segmentation maps sized at $1024 \times 1024$. Similar to other two datasets, we train the learnable parameters for 100 epochs with a batch size of 4.

**CMU-MOSI and CMU-MOSEI Datasets:** We used Multimodal Transformer (MulT) from Tsai et al. (2019) as the base model. Preprocessed datasets and all the configurations are available on the repository[6]. First we trained the multimodal transformer (MulT) model on all the available modalities and then adapted the pretrained model for different modalities. The hyperparameters for the experiments are shown in Table 6.

## C    NUMBER OF LEARNABLE PARAMETERS

We report the number of learnable parameters for different parameter-efficient adaptation techniques (for multimodal segmentation) in Table 7. We insert scale and shift layers after each linear, convolutional and norm (both batch norm and layer norm) layers. The number of learnable parameter varies with the size of the backbone. We used MiT-B4 as the backbone while counting these learnable parameters. Scale and shift adds only 0.789M learnable parameters which is less than 0.7% of the total model parameters. Despite this very few parameters, it improves performance significantly in different missing modality scenarios. For this study we mainly focused on improving missing modality robustness and did not try to optimize the number of learnable parameters. We will leave that part for future studies.

---

[5]https://github.com/charlesCXK/RGBD_Semantic_Segmentation_PyTorch
[6]https://github.com/thuiar/MMSA

**Table 8:** Performance comparison with parameter efficient model adaptation techniques on CMNeXt model for MFNet dataset. Average column indicates average performance when one of the two modalities gets missing. Mean accuracy, F1 score and % mIoU are shown for all the experiments.

| Methods | RGB | | | Thermal | | | Average | | |
|---|---|---|---|---|---|---|---|---|---|
| | mAcc | F1 | % mIoU | mAcc | F1 | % mIoU | mAcc | F1 | % mIoU |
| Pretrained | 60.74 | 66.91 | 53.71 | 38.18 | 45.11 | 35.48 | 49.46 | 56.01 | 44.60 |
| Dedicated | 66.28 | 68.22 | 55.86 | 68.35 | 65.29 | 53.34 | 67.32 | 66.76 | 54.60 |
| Scale Only | 67.09 | 68.03 | 54.77 | 64.00 | 60.92 | 49.23 | 65.55 | 64.48 | 52.00 |
| Shift Only | 65.82 | 67.42 | 54.57 | 59.77 | 60.54 | 48.96 | 62.80 | 63.98 | 51.77 |
| BitFit | 66.49 | 67.40 | 54.39 | 61.06 | 60.59 | 49.07 | 63.78 | 64.00 | 51.73 |
| LoRA | 66.44 | 67.32 | 54.19 | 57.10 | 59.04 | 47.45 | 61.77 | 63.18 | 50.82 |
| Norm | 66.43 | 67.07 | 54.65 | 57.55 | 59.22 | 47.49 | 61.99 | 63.15 | 51.07 |
| Scale and Shift | **67.18** | **68.04** | **55.22** | **66.70** | **62.64** | **50.89** | **66.94** | **65.34** | **53.06** |

**Table 9:** Performance comparison with parameter efficient model adaptation techniques on CMNeXt model for NYUDv2 dataset. Average column indicates average performance when one of the two modalities gets missing. Mean accuracy, F1 score and % mIoU are shown for all the experiments.

| Methods | RGB | | | Depth | | | Average | | |
|---|---|---|---|---|---|---|---|---|---|
| | mAcc | F1 | % mIoU | mAcc | F1 | % mIoU | mAcc | F1 | % mIoU |
| Pretrained | 64.10 | 65.70 | 51.19 | 8.30 | 7.95 | 5.26 | 36.20 | 36.83 | 28.23 |
| Dedicated | 66.00 | 66.62 | 52.18 | 44.80 | 46.79 | 33.49 | 55.40 | 56.71 | 42.84 |
| Scale Only | **68.18** | **67.38** | 53.04 | 51.54 | 49.88 | 36.12 | 59.86 | 58.63 | 44.58 |
| Shift Only | 67.54 | 67.35 | 53.04 | 50.30 | 49.76 | 36.25 | 58.92 | 58.56 | 44.65 |
| BitFit | 67.31 | 67.33 | **53.09** | 50.68 | 50.27 | 36.64 | 59.00 | 58.80 | **44.87** |
| LoRA | 66.67 | 67.14 | 52.87 | 49.34 | 48.66 | 34.97 | 58.01 | 57.90 | 43.92 |
| Norm | 67.18 | 67.34 | 53.05 | 48.74 | 48.06 | 34.73 | 57.96 | 57.70 | 43.89 |
| Scale and Shift | 67.96 | 67.18 | 52.82 | **52.42** | **50.60** | **36.72** | **60.19** | **58.89** | 44.77 |

# D   COMPARISON OF PARAMETER-EFFICIENT ADAPTATION METHODS

We performed a detailed performance comparison with other parameter efficient methods for the three segmentation datasets. The results are summarized in Table 8 for RGB-thermal segmentation on MFNet dataset, Table 9 for RGB-depth segmentation on NYUDv2 dataset and Table 10 for multimodal material segmentation on MCubeS dataset. For each method, we take a model trained on all the available modalities. Then we freeze the pretrained weights and tune the learnable parameters for the corresponding adaption method. We have shown mean accuracy, F1 score and % mIoU for each experiment.

## D.1   COMPARISON FOR RGB-THERMAL SEMANTIC SEGMENTATION

Table 8 summarizes the results on MFNet dataset when the base CMNeXt model is adapted with other parameter efficient model adaptation techniques. Experiments show that scale and shift shows the best performance in all three matrices compared to all other methods. It shows a significant improvement of +8.46% in mIoU, +9.33% in F1 score and +17.48% in mean accuracy on an average over the pretrained model. The average performance is also close to dedicatedly trained models.

## D.2   COMPARISON FOR RGB-DEPTH SEMANTIC SEGMENTATION

Similar trend is observed fro RGB-depth semantic segmentation on NYUDv2 dataset as shown in Table 9. Scale only and BitFit adapted models show slightly better performance for some of the matrices. But in most of the cases scale and shift adapted model performs better. For all the matrices, scale and shift shows a significant improvement of +16.54% in mIoU, +22.05% in F1 score and +23.99% in mean accuracy over the pretrained model on an average and consistently outperforms dedicated training.

**Table 10:** Performance comparison with different parameter efficient model adaptation techniques on CMNeXt model for MCubeS dataset. Average column indicates the average performance. Mean accuracy, F1 score and % mIoU are shown for all the experiments.

| Methods | RGB | | | RGB-AoLP | | | RGB-AoLP-DoLP | | | Average | | |
|---|---|---|---|---|---|---|---|---|---|---|---|---|
| | mAcc | F1 | % mIoU | mAcc | F1 | % mIoU | mAcc | F1 | % mIoU | mAcc | F1 | % mIoU |
| Pretrained | 51.63 | 55.91 | 42.32 | 58.66 | 62.00 | 48.81 | 60.06 | 62.43 | 49.06 | 56.78 | 60.11 | 46.73 |
| Dedicated | 57.70 | 60.95 | 48.16 | 57.56 | 61.17 | 48.42 | 59.12 | 61.91 | 49.48 | 58.13 | 61.34 | 48.69 |
| Scale Only | 59.64 | 63.06 | 50.16 | 60.28 | 63.55 | 50.55 | 60.96 | **64.14** | **51.13** | 60.29 | 63.58 | 50.61 |
| Shift Only | 59.82 | 63.17 | 50.13 | 60.10 | 63.36 | 50.40 | 60.61 | 63.78 | 50.86 | 60.18 | 63.44 | 50.46 |
| BitFit | 59.98 | 63.24 | 50.19 | 60.12 | 63.52 | 50.57 | 60.84 | 64.03 | 51.07 | 60.31 | 63.60 | 50.61 |
| LoRA | 59.08 | 62.50 | 49.59 | 59.81 | 63.05 | 50.07 | 60.69 | 63.84 | 50.80 | 59.86 | 63.13 | 50.15 |
| Norm | 59.57 | 62.89 | 49.95 | 60.22 | 63.49 | 50.51 | **60.98** | 64.08 | 51.07 | 60.26 | 63.49 | 50.51 |
| Scale and Shift | **60.23** | **63.41** | **50.43** | **60.40** | **63.59** | **50.62** | 60.94 | 64.04 | 51.11 | **60.52** | **63.68** | **50.72** |

## D.3 COMPARISON FOR MULTIMODAL MATERIAL SEGMENTATION

We show comparison with parameter efficient model adaptation techniques on MCubeS dataset in Table 10. Scale and shift outperforms all other methods in most of the matrices for all input combinations. It also shows an improvement of +3.99% in mIoU, +3.57% in F1 score and +3.74% in mean accuracy on an average over the pretrained model. Furthermore, Scale and shift also outperforms dedicated training for all input modality combinations. These experiments corroborate the fact that scale and shift provides better model adaption for different missing modality scenarios.

## E PER CLASS IoU COMPARISON

To further analyze how the adaption is helping the model improve overall semantic and material segmentation performance, we conduct a per-class % intersection over union (IoU) analysis on the pretrained and adapted models. Table 11 summarizes the results. We show the per class % IoU comparison for different missing modality situations on MFNet dataset on Table 11a. From the table we can see that when RGB is available and thermal is missing, the adaptation helps improve performance for most of the classes. Though we see some performance drop for bike (-0.74%) and guardrail (-5.25%) classes, the rest of the classes have better % IoU than the pretrained model. Bump (+6.26%), person (+4.42%), and curve (+4.14%) classes show greater improvement after adaptation. When thermal is available and RGB is missing, adaptation improves performance for all the classes. Among the classes, bike (+39.76%), car (+27.59%), car stop (+24.04%), color cone (+19.37%) and bump (+13.91%) are showing impressive performance improvement over the pretrained model.

Results for MCubeS dataset is shown on Table 11b. Here A, D, and N stand for angle of linear polarization (AoLP), degree of linear polarization (DoLP) and near-infrared (NIR) respectively. Experiments show that when only RGB is available and the rest of the modalities are missing, the adapted model performs better in detecting all the 20 classes present in the dataset. Gravel (36.2%), asphalt (16.1%), rubber (15.2%), wood (12.9%) and sky (10.2%) are some of the classes who show the most performance boost after adaptation. In other input combinations, most of the classes see performance improvement compared to the pretrained model. Though we see some performance drop in a few classes, most of the classes show improvement in % IoU which leads to the overall performance improvement after adaption.

## F COMPARISON WITH AVAILABLE MODALITY DUPLICATION METHOD FOR MISSING MODALITIES

Apart from replacing the missing modalities with zeros, we have also performed some experiments on utilizing available modalities to compensate for the missing modalities. We compare the performance of duplicating available modalities and replacing zeros for missing modalities with our adaptation method in Table 12. The results show that duplicating available modalities is inferior to the adapted model (and in some cases worse than replacing zeros for missing modalities). For MFNet and NYUDv2 datasets, we replaced the missing modality with a copy of the available modality. For

**Table 11:** Per class % IoU comparison between pretrained and adapted CMNeXt model on MFNet and MCubeS datasets. Adapted model show better performance for most of the classes leading to overall performance improvement. Here A, D and N stand for Angle of Linear Polarization (AoLP), Degree of Linear Polarization (DoLP) and Near-Infrared (NIR) respectively.

**(a)** Per class % IoU of pretrained and adapted CMNeXt model on MFNet dataset.

| Modalities | Methods | Unlabeled | Car | Person | Bike | Curve | Car_Stop | Guardrail | Color_Cone | Bump | Mean |
|---|---|---|---|---|---|---|---|---|---|---|---|
| RGB-Thermal | CMNeXt | 98.31 | 90.27 | 74.52 | 64.52 | 46.64 | 39.19 | 15.09 | 52.56 | 59.79 | 60.10 |
| RGB | Pretrained | **97.79** | 87.62 | 51.13 | **61.94** | 30.05 | 39.36 | **21.04** | 45.55 | 48.95 | 53.71 |
| | Adapted | **97.79** | 88.06 | 55.55 | 61.20 | 34.19 | 40.52 | 15.78 | 48.67 | 55.21 | **55.22** |
| Thermal | Pretrained | 95.97 | 55.24 | 68.47 | 9.27 | 31.85 | 2.75 | 0.0 | 16.87 | 38.92 | 35.48 |
| | Adapted | **97.46** | **82.83** | **70.12** | **49.03** | **40.89** | **26.79** | 1.84 | **36.24** | **52.83** | **50.89** |

**(b)** Per class % IoU of pretrained and adapted CMNeXt model on MCubeS dataset.

| Modalities | Methods | Asphalt | Concrete | Metal | Road_Marking | Fabric | Glass | Plaster | Plastic | Rubber | Sand | Gravel | Ceramic | Cobblestone | Brick | Grass | Wood | Leaf | Water | Human | Sky | Mean |
|---|---|---|---|---|---|---|---|---|---|---|---|---|---|---|---|---|---|---|---|---|---|---|
| RGB-A-D-N | CMNeXt | 84.4 | 44.9 | 53.9 | 74.6 | 32.1 | 54.0 | 0.8 | 28.7 | 29.8 | 67.0 | 66.2 | 27.7 | 68.5 | 42.8 | 58.7 | 49.7 | 75.3 | 55.6 | 19.1 | 96.52 | 51.5 |
| RGB | Pretrained | 69.7 | 39.2 | 47.6 | 67.3 | 26.9 | 44.6 | 0.2 | 20.9 | 15.2 | 61.8 | 36.7 | 19.1 | 67.2 | 36.0 | 49.5 | 36.1 | 71.6 | 36.1 | 14.7 | 86.3 | 42.3 |
| | Adapted | **85.8** | 43.7 | 52.6 | 73.8 | 27.9 | 51.0 | 0.8 | 24.2 | 30.4 | 67.8 | 72.9 | 27.1 | 68.1 | 42.9 | 57.6 | 49.0 | 74.9 | 43.4 | 18.3 | 96.5 | 50.4 |
| RGB-A | Pretrained | 83.2 | 43.3 | 50.7 | 72.6 | 26.4 | 51.9 | 0.2 | **28.1** | 22.2 | 67.1 | 63.4 | 22.7 | **67.5** | 40.6 | 54.4 | 44.9 | 73.9 | 44.8 | **21.8** | 96.0 | 48.8 |
| | Adapted | 84.4 | 45.4 | 53.8 | 74.5 | 30.4 | 53.2 | 0.6 | 26.9 | 28.8 | 69.0 | 69.3 | 24.8 | 67.5 | 43.2 | 58.4 | 48.2 | 75.1 | 48.1 | 14.4 | 96.4 | 50.6 |
| RGB-A-D | Pretrained | **84.5** | 41.2 | 46.7 | 72.8 | 25.2 | 51.6 | 0.3 | 26.1 | 28.8 | 66.7 | 65.6 | **26.0** | 66.5 | 40.4 | 50.0 | 45.1 | 72.7 | 49.4 | **25.6** | 96.3 | 49.1 |
| | Adapted | 84.1 | 45.6 | 54.1 | 74.6 | 30.5 | 54.2 | 0.6 | 28.1 | 30.1 | 69.0 | 67.6 | 25.9 | 67.8 | 43.8 | 58.0 | 49.1 | 75.0 | 53.7 | 13.7 | 96.5 | 51.1 |

**Table 12:** Performance of Pretrained model with zero filling, Pretrained model with available modality duplication, and Adapted model for missing modalities. CMNeXt is used as the base model for all the datasets. HHA-encoded images were used instead of raw depth maps. **Bold** letters represent best results.

| Dataset | Input | Missing | Pretrained (Zeros for missing) | Pretrained (Duplicate for missing) | Adapted (Ours) |
|---|---|---|---|---|---|
| MFNet | RGB-Thermal | - | 60.10 | - | - |
| | RGB | Thermal | 53.71 | 52.33 | **55.22** |
| | Thermal | RGB | 35.48 | 44.43 | **50.89** |
| NYUDv2 | RGB-Depth | - | 56.30 | | |
| | RGB | Depth | 51.19 | 46.19 | **52.82** |
| | Depth | RGB | 5.26 | 13.94 | **36.72** |
| MCubeS | RGB-AoLP-DoLP-NIR | - | 51.54 | - | - |
| | RGB-AoLP-DoLP | NIR | 49.06 | 49.93 | **51.11** |
| | RGB-AoLP | DoLP-NIR | 48.81 | 49.23 | **50.62** |
| | RGB | AoLP-DoLP-NIR | 42.32 | 48.96 | **50.43** |

the MCubeS dataset, we replaced the missing modalities with a copy of RGB modality. In all the combinations, the adapted model works much better than the modality duplication method.

## G  EXAMPLES OF PREDICTED SEGMENTATION MAPS

We show the predicted segmentation maps from the pretrained and adapted models in Figure 3. For each dataset, we show the input images, predictions from the base CMNeXt model when all the modalities are available, predictions from the adapted and pretrained models for different missing modality scenarios. For brevity, we only show RGB input images for MCubeS dataset. A, D and N stand for angle of linear polarization (AoLP), degree of linear polarization (DoLP) and near-infrared

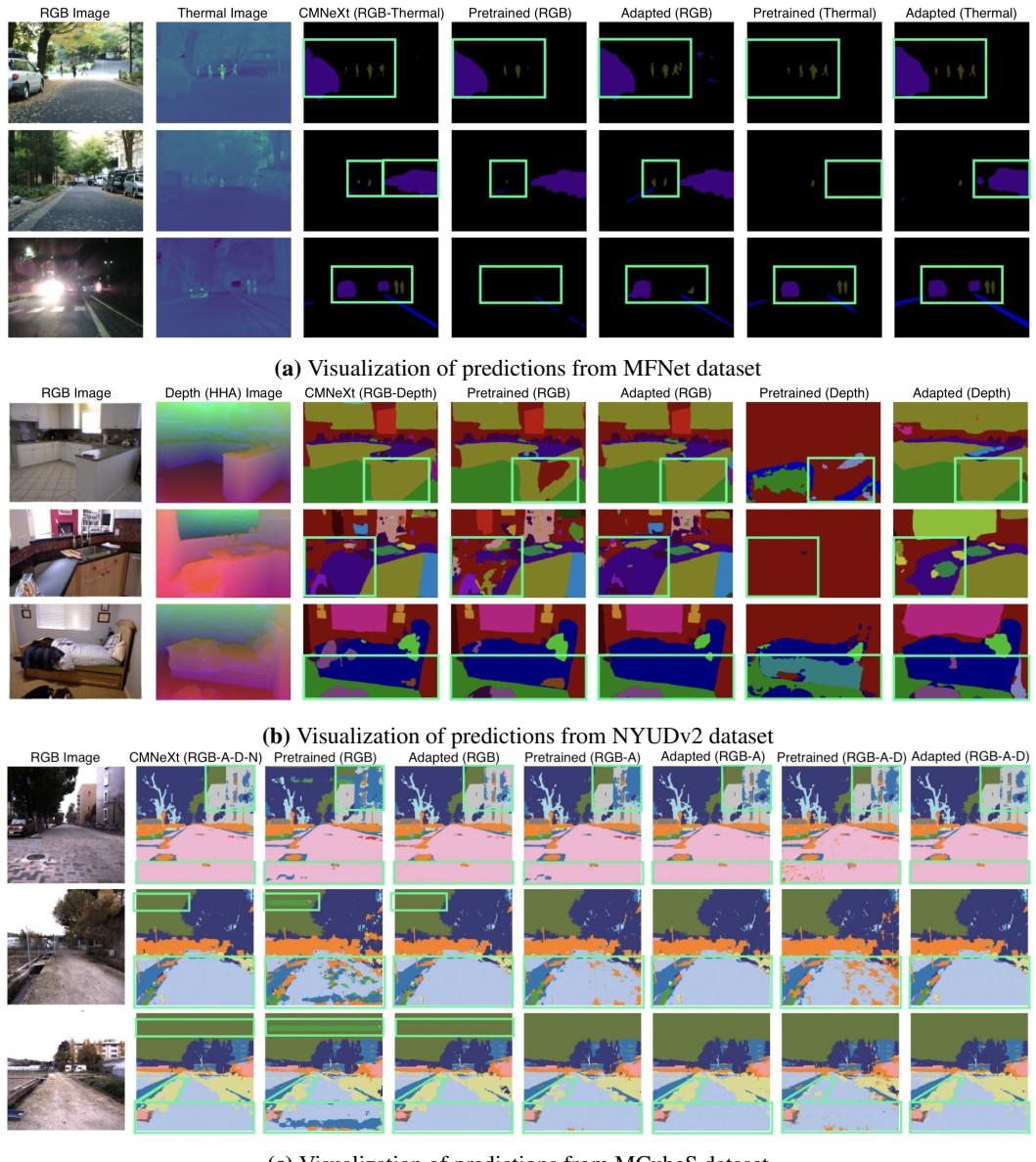

**(a)** Visualization of predictions from MFNet dataset

**(b)** Visualization of predictions from NYUDv2 dataset

**(c)** Visualization of predictions from MCubeS dataset

**Figure 3:** Visualization of predicted segmentation maps for pretrained and adapted models on MFNet and NYUDv2 datasets for multimodal semantic segmentation and MCubeS dataset for multimodal material segmentation. Only RGB images are shown from MCubeS dataset for brevity. CMNeXt column shows the predictions when all the modalities are available. Segmentation quality improves significantly after model adaptation for all the input modality combinations. A, D and N stand for angle of linear polarization, degree of linear polarization and near-infrared respectively.

(NIR) respectively. Modalities that are available during testing are shown in parenthesis while other modalities are missing.

For MFNet dataset, Figure 3a shows that when only RGB is available, the pretrained model performs very poorly in detecting humans. On the other hand, if only thermal is available, the pretrained model can not detect cars very accurately. But the adapted model can detect both humans and cars more accurately in both of the scenarios. In all the cases, the predictions form the adapted model is closer to the predictions of the base CMNeXt model when all the modalities are available.

Predictions from NYUDv2 dataset is shown on Figure 3b. We can see that the adapted model can identify bed, furniture and other classes more accurately than the pretrained model for different

**Table 13:** Comparison of our adaptation technique with existing methods for multimodal sentiment analysis.

| Datasets | Methods | Audio | | Visual | | Audio-Visual | | Average | |
|---|---|---|---|---|---|---|---|---|---|
| | | ACC | F1 | ACC | F1 | ACC | F1 | ACC | F1 |
| CMU -MOSI | MulT (Tsai et al. (2019)) | 48.31 | 40.98 | 52.44 | 51.77 | 48.93 | 41.95 | 49.89 | 44.90 |
| | MFN (Zadeh et al. (2018)) | 56.86 | 44.81 | 55.95 | 42.94 | 56.86 | 51.07 | 56.56 | 46.27 |
| | TFN (Zadeh et al. (2017)) | 42.23 | 25.07 | 42.38 | 25.40 | 42.23 | 25.07 | 42.28 | 25.18 |
| | BERT_MAG (Rahman et al. (2020)) | **57.77** | 42.31 | **57.77** | 42.31 | **57.77** | 42.31 | **57.77** | 42.31 |
| | LMF (Liu et al. (2018)) | 42.23 | 25.07 | 43.14 | 27.54 | 43.29 | 27.61 | 42.89 | 26.74 |
| | Adapted (Ours) | 50.00 | **46.71** | 54.88 | **54.39** | 55.49 | **53.96** | 53.46 | **51.69** |
| CMU - MOSEI | MulT (Tsai et al. (2019)) | 37.15 | 20.12 | 38.28 | 23.70 | 41.91 | 32.78 | 39.11 | 25.53 |
| | MFN (Zadeh et al. (2018)) | 58.48 | **58.31** | 60.35 | 59.48 | 59.74 | 60.37 | 59.52 | **59.39** |
| | TFN (Zadeh et al. (2017)) | 37.15 | 20.12 | 37.15 | 20.12 | 37.15 | 20.12 | 37.15 | 20.12 |
| | BERT_MAG (Rahman et al. (2020)) | 62.83 | 48.50 | 61.39 | 49.70 | 62.83 | 48.51 | 62.35 | 48.90 |
| | LMF (Liu et al. (2018)) | 42.38 | 34.48 | 57.15 | 57.85 | 55.94 | 56.63 | 51.82 | 49.65 |
| | Adapted (Ours) | **62.85** | 55.55 | **62.49** | 60.00 | 63.32 | 60.69 | **62.89** | 58.75 |

missing modality scenarios. The pretrained model performs very poorly when only depth is available and RGB is missing. But detection accuracy improves significantly after model adaptation. For MCubeS dataset, as seen in Figure 3c, predictions from the pretrained model shows artifacts when detecting different materials. On the other hand, the adapted model is showing more accuracy in detecting sky, cobblestone, sand and brick. For all the three datasets, the predictions from the adapted model is more accurate and closer to the all modality predictions of the base CMNeXt model.

# H   COMPARISON WITH EXISTING METHODS FOR MULTIMODAL SENTIMENT ANALYSIS

We also compare our adaptation method with existing methods for multimodal sentiment analysis and show the results in Table 13. All the experiments are done with the same settings described in Section B utilizing the same codebase for the model implementations. The table shows that for CMU-MOSI dataset, BERT_MAG works better in terms of accuracy but our adaptation method works better in terms of F1 score. One thing to mention is that BERT_MAG uses a pretrained BERT model and finetunes it on the dataset but we are not using any pretraining. For CMU-MOSEI, our adaptation method works better for most of the cases.

