# OpenReview forum: "Robust Multimodal Learning with Missing Modalities via Parameter-Efficient Adaptation"
_ICLR.cc/2024/Conference — Submitted to ICLR 2024_

### Official Review · Reviewer_Gaao · 2023-10-30

**Soundness:** 3 good
**Presentation:** 3 good
**Contribution:** 3 good
**Rating:** 5
**Confidence:** 3

**Summary:**

This paper introduces parameter-efficient adaptation to make multi-modal segmentation more robust in the face of missing modalities, and it compares various methods, demonstrating promising results.

**Strengths:**

1, The method in this paper utilizes parameter-efficient fine-tuning to make the model adapt to modality loss, which I find relatively straightforward. The performance is also quite good.

2, The paper is well-written and relatively easy to read.

**Weaknesses:**

1, The method proposed in this paper may not be directly applicable to scenarios where the training set has missing modalities.

2, The method proposed in this paper lacks extensive comparisons with other methods~(for example, [1,2,3]) in the context of multimodal classification tasks, which significantly undermines the persuasiveness of Table 5.


[1] Are Multimodal Transformers Robust to Missing Modality?

[2] Multi-modal Learning with Missing Modality via Shared-Specific Feature Modelling

[3] What makes for robust multi-modal models in the face of missing modalities?

**Questions:**

See weakness.

---

> ### Author Response · Authors · 2023-11-16
> **Response to Reviewer Gaao**
>
> Thank you very much for your review. We thank you for acknowledging the strengths, simplicity and performance of our method. We respond to your other comments below.
>
> 1. **The method proposed in this paper may not be directly applicable to scenarios where the training set has missing modalities.** \
> This is a good point. Indeed if a modality is completely missing from the training set, then our method (and almost all existing methods) cannot compensate for that. However, if the training set has some samples missing for some modalities, we can pretrain a network with the available modalities and then perform parameter-efficient network adaptation to compensate for the missing modalities at the test time. Since our approach is mainly focused on adapting a pretrained model to different missing modality situations, we can use the same adaptation procedure using a different base network. The performance in this case will probably depend on the quality of the base model pretrained with missing samples from different modalities.
> ######
> 2. **The method proposed in this paper lacks extensive comparisons with other methods\~(for example, \[1,2,3]) in the context of multimodal classification tasks, which significantly undermines the persuasiveness of Table 5.** \
> Our work focuses on parameter efficient adaptation techniques for existing multimodal models. The main goal is to adapt already pretrained models to different missing modality scenarios. While we cited the first two papers (\[1, 2]) in our work, we did not compare with them for the following reasons:\
> \
> **Paper \[1]** suggested multitask learning and search for optimal fusion strategy to enhance robustness (mainly for two modalities). Extending the methods for masking out attention for more than two modalities and for tasks that rely on all the tokens is not obvious. Furthermore, the paper did not provide a codebase or results for the datasets we used in our experiments; therefore, we could not compare with them.\
> \
> **Paper \[2]** suggested Shared Specific Feature Modelling (ShaSpec), which is based on modeling and fusing shared and specific features. The paper focused on medical imaging (brain tumor segmentation) and classification. They did not provide any baseline for the datasets and tasks that we used in our work. That is why we did not compare our results with the paper.\
> \
> **Paper \[3] -** This paper was submitted to arxiv 12 days after the ICLR submission deadline. For this reason, we could not cite or compare with this paper. We will be happy to cite the paper in the revised version. The paper did not provide code or baseline results for the datasets and tasks that we used in our paper. For these reasons, we are unable to provide a comparison.

---

> > ### Author Response · Authors · 2023-11-22
> >
> > Dear Reviewer,\
> > We hope you had a chance to read our response to your questions. We appreciate your feedback. Please let us know if you have any additional question that we can answer today. Thank you for your kind attention.

---

### Official Review · Reviewer_ibUc · 2023-10-31

**Soundness:** 2 fair
**Presentation:** 3 good
**Contribution:** 2 fair
**Rating:** 5
**Confidence:** 3

**Summary:**

The paper tackles the problem of missing modalities at test time for multimodal learning. The motivation is to introduce small number of (new) parameters as much as possible during adpatation of a pre-trained model. The paper proposes parameter-efficient adpatation of multimodal networks.  Three different parameter-efficient adaptation techniques, including low-rank adpatation, scaling and shifting of features, and BitFit, have been investigated towards learning a model robust to missing modalities. Given a pre-trained network, in order to adapt this model to a subset of modalities, small set of parameters (as a layer) are introduced after each layer of the network. During adaptation to this subset of modality combination, only these small set of parameters are trained. Experiments have been performed with two different base networks for the task of multimodal semantic segmentation and multimodal sentiment analyses. Results claim to achieve better performance than competing robust methods and often surpass dedicated models.

**Strengths:**

- Development of multimodal learning algorithms robust to missing modalities is crucial since observations from some modality could go often be missing due to sensor/hardware failure.

- The proposed method is effective and importantly efficient as it introduces a very small set of parameters towards making a pre-trained model robust to missing modalities.

- The paper is mostly well-written and easier to read.

- Experiments have been performed on two different tasks of multimodal semantic segmentation and multimodal sentiment analyses. Results claim to surpass the existing methods and (often) dedicated methods when the modalities are missing at test time.

**Weaknesses:**

- Although the idea is simple and effective for achieving parameter-efficient adaptation of models, the application of existing techniques (i.e. low rank adaptation, scale and shift transformations, and BitFit) to adapt a pre-trained model on a subset of modalities seems low from novelty aspect.


- There is no insight and/or analyses into how the proposed paramater-efficient is helpful towards making the model robust to missing modalities. It is important to develop the grounding of the proposed method since only showing improvements over the baseline with empirical results doesn't help much towards this.

- The paper doesn’t mention any clear justification on why SSA was chosen over the LoRA and BitFit to report the results in the paper.

- Comparison is missing with any of the existing method for the task of multimodal sentiment analyses.

- Why CMU-MOSI and CMU-MSEI datasets were chosen to evaluate and report the performance of the method?

**Questions:**

- There is no study that fairly compares the parameter efficiency of the proposed method with competing methods?

- What are the possible reasons for relatively lower performance of other adaptation strategies, compared to SSA, in Table 4? Some explanation would be helpful to better understand their comparison in missing modalities scenario.

**Details Of Ethics Concerns:**

No ethics concerns.

---

> ### Author Response · Authors · 2023-11-16
> **Response to Reviewer ibUc (Part 1/3)**
>
> Thank you very much for your insightful and detailed review. Thank you for nicely summarizing the strengths and contributions of our paper. We respond to your questions and other comments below.
>
> 1. **Although the idea is simple and effective for achieving parameter-efficient adaptation of models, the application of existing techniques (i.e. low rank adaptation, scale and shift transformations, and BitFit) to adapt a pre-trained model on a subset of modalities seems low from a novelty aspect.**\
> We agree with you that the idea is simple and effective. Low rank adaptation, BitFit and Scale and Shift are originally designed for efficient model finetuning and transfer learning. We show that they can also be applied to adapt models to different missing modality scenarios.\
>    1\) To the best of our knowledge, we are the first to show that parameter efficient adaptation can be used to enhance performance in different missing modality scenarios. While it may be simple, we hope this will be informative and useful for the broader community.\
>    2\) We provide a generic framework for missing modality adaptation that can be applied to a wide range of multimodal datasets, tasks, and models.\
>    3\) The adapted model can easily switch to different network states based on the available modalities with minimal latency, computational, or memory overhead.
> ######
> 2. **There is no insight and/or analyses into how the proposed parameter-efficient is helpful towards making the model robust to missing modalities. It is important to develop the grounding of the proposed method since only showing improvements over the baseline with empirical results doesn't help much towards this.**\
> Our main motivation and insight for the proposed approach is to adapt the functional representation of the network according to the available modality combination. For instance, if one of the modalities is missing, we want to modify the feature representations and fusion of other modalities. We could do this by changing the entire network for every modality combination, which is infeasible due to computational and storage cost. In this paper, we perform parameter-efficient adaptation to achieve the same goal. A rigorous analysis of how the network adaptation changes the functional representation of the network will certainly be great, but it is a nontrivial task and beyond the scope of one paper.
> ######
> 3. **The paper doesn’t mention any clear justification on why SSA was chosen over the LoRA and BitFit to report the results in the paper.**\
> We have discussed this in the second paragraph of Section 3.3. We primarily selected the SSF technique because of its simplicity and effectiveness. SSF has several benefits over other adaptation methods:\
>    1\) The parameters (γ, β) are independent of the input features, which makes SSF applicable to diverse tasks and input modality combinations. \
>    2\) We can easily insert SSF layers in the existing model without changing the model architecture.\
>    3\) We can easily switch/select the corresponding SSF parameters for a given input modality combination.\
>    4\) SSF works relatively better than LoRa, BitFit and Norm in most of the scenarios as shown in Table 4.

---

> > ### Author Response · Authors · 2023-11-16
> > **Response to Reviewer ibUc (Part 2/3)**
> >
> > 4. **Comparison is missing with any of the existing methods for the task of multimodal sentiment analysis.**\
> > Thank you very much for pointing this out. We are including the comparison with existing methods for multimodal sentiment analysis in the table below. All the results are generated using the codebase available at: [**https://github.com/thuiar/MMSA**](https://github.com/thuiar/MMSA) with the same settings that we used for our experiments.
> >
> >    |             |                |           |           |           |           |              |           |           |           |
> >    | :--------- | :------------ | :-------: | :-------: | :-------: | :-------: | :----------: | :-------: | :-------: | :-------: |
> >    |   Datasets  |     Methods    |    Audio  ||   Visual             || Audio-Visual          ||  Average         ||
> >    |             |                |    ACC    |     F1    |    ACC    |     F1    |      ACC     |     F1    |    ACC    |     F1    |
> >    |  CMU -MOSI  |    MulT \[1]   |   48.31   |   40.98   |   52.44   |   51.77   |     48.93    |   41.95   |   49.89   |   44.90   |
> >    |       |    MFN \[2]    |   56.86   |   44.81   |   55.95   |   42.94   |     56.86    |   51.07   |   56.56   |   46.27   |
> >    |       |    TFN \[3]    |   42.23   |   25.07   |   42.38   |   25.40   |     42.23    |   25.07   |   42.28   |   25.18   |
> >    |       | BERT\_MAG \[4] | **57.77** |   42.31   | **57.77** |   42.31   |   **57.77**  |   42.31   | **57.77** |   42.31   |
> >    |       |    LMF \[5]    |   42.23   |   25.07   |   43.14   |   27.54   |     43.29    |   27.61   |   42.89   |   26.74   |
> >    |       |      Ours      |   50.00   | **46.71** |   54.88   | **54.39** |     55.49    | **53.96** |   53.46   | **51.69** |
> >    | CMU - MOSEI |    MulT \[1]   |   37.15   |   20.12   |   38.28   |   23.70   |     41.91    |   32.78   |   39.11   |   25.53   |
> >    |       |    MFN \[2]    |   58.48   | **58.31** |   60.35   |   59.48   |     59.74    |   60.37   |   59.52   | **59.39** |
> >    |       |    TFN \[3]    |   37.15   |   20.12   |   37.15   |   20.12   |     37.15    |   20.12   |   37.15   |   20.12   |
> >    |       | BERT\_MAG \[4] |   62.83   |   48.50   |   61.39   |   49.70   |     62.83    |   48.51   |   62.35   |   48.90   |
> >    |       |    LMF \[5]    |   42.38   |   34.48   |   57.15   |   57.85   |     55.94    |   56.63   |   51.82   |   49.65   |
> >    |       |      Ours      | **62.85** |   55.55   | **62.49** | **60.00** |   **63.32**  | **60.69** | **62.89** |   58.75   |
> >
> >    From the table above, we can see that for the CMU-MOSI dataset, BERT\_MAG works better in terms of accuracy but our method works better in terms of F1 score. One thing to mention is that BERT\_MAG uses a pre-trained BERT model and finetunes it on the dataset but we are not using any pre-training. For CMU-MOSEI, our adaptation method works better for most of the cases.
> > ######
> > 5. **Why were CMU-MOSI and CMU-MSEI datasets chosen to evaluate and report the performance of the method?**\
> > We wanted to evaluate our method for diverse modality combinations. We have shown results for RGB, depth and thermal modalities for multimodal semantic segmentation. Multimodal material segmentation uses RGB, thermal, angle of linear polarization and degree of linear polarization. Apart from these modalities, we also wanted to test our approach on other modalities including video, audio and text. CMU-MOSI and CMU-MSEI datasets have those modalities which corroborate the fact that our approach can be applied to a diverse number of modalities.
> > ****
> > \[1] Yao-Hung Hubert Tsai, Shaojie Bai, Paul Pu Liang, J. Zico Kolter, Louis-Philippe Morency, Ruslan Salakhutdinov; “Multimodal Transformer for Unaligned Multimodal Language Sequences”, Proceedings of the 57th Annual Meeting of the Association for Computational Linguistics (Volume 1: Long Papers), 2019
> >
> > \[2] Amir Zadeh, Paul Pu Liang, Navonil Mazumder, Soujanya Poria, Erik Cambria, Louis-Philippe Morency; “Memory Fusion Network for Multi-view Sequential Learning”, Proceedings of the Thirty-Second {AAAI} Conference on Artificial Intelligence, 2018
> >
> > \[3] Amir Zadeh, Minghai Chen, Soujanya Poria, Erik Cambria, Louis-Philippe Morency; “Tensor Fusion Network for Multimodal Sentiment Analysis”, Proceedings of the 2017 Conference on Empirical Methods in Natural Language Processing, 2017
> >
> > \[4] Wasifur Rahman, Md. Kamrul Hasan, Sangwu Lee, Amir Zadeh, Chengfeng Mao, Louis-Philippe Morency, Ehsan Hoque; "Integrating Multimodal Information in Large Pretrained Transformers", Proceedings of the 58th Annual Meeting of the Association for Computational Linguistics, 2020
> >
> > \[5] Zhun Liu, Ying Shen, Varun Bharadhwaj Lakshminarasimhan, Paul Pu Liang, AmirAli Bagher Zadeh, Louis-Philippe Morency; "Efficient Low-rank Multimodal Fusion With Modality-Specific Factors", Proceedings of the 56th Annual Meeting of the Association for Computational Linguistics (Volume 1: Long Papers), 2018

---

> ### Author Response · Authors · 2023-11-16
> **Response to Reviewer ibUc (Part 3/3)**
>
> 6. **There is no study that fairly compares the parameter efficiency of the proposed method with competing methods?**\
> Table 4 provides comparison of different parameter-efficient adaptation methods that can be used in our proposed framework.
> We compared the performance of our adaptation-based method with existing robust methods in Table 2 and 3.  Since these methods are primarily designed to be robust to missing modalities, their emphasis is not necessarily to learn parameter-efficient networks. For this reason, we felt a comparison in terms of parameter efficiency would be unnecessary or unfair. We are listing the total number of parameters for some of these models here for completeness.
>
>    |                             |                                                        |
>    | --------------------------- | ------------------------------------------------------ |
>    | Method                      | Total Number of Parameters (M)                         |
>    | CEN                         | 118.2                                                  |
>    | MDRNet                      | 64.60                                                  |
>    | RTFNet (ResNet-152)         | 254.51                                                 |
>    | SAGate                      | 110.85                                                 |
>    | AsymFussion (ResNet-101)    | 118.2                                                  |
>    | Ours (Adapted CMNext Model) | 117.3 (Base Model: 116.56M, Adaptation Layers: 0.789M) |
>    |  |  |
> ######
> 7. **What are the possible reasons for relatively lower performance of other adaptation strategies, compared to SSA, in Table 4? Some explanation would be helpful to better understand their comparison in the missing modalities scenario.**\
> SSF learns parameters ($\gamma$, $\beta$) for scaling and shifting the intermediate features. We can consider BitFit as a subset of SSF which only shifts the intermediate features. LoRA is only applied to the attention layers ($W_q$ and $W_v$) following the original paper and the number of parameters for LoRA is less than SSF. Norm only optimizes the norm layers while SSF is applied after every linear, convolutional and norm layer. Which gives SSF more advantage over other methods to modulate the intermediate features and perform better on downstream tasks.

---

> > ### Comment · Reviewer_ibUc · 2023-11-22
> >
> > I thank authors for submitting the responses to all my comments. Particularly, i appreciate the new comparisons provided on multimodal sentiment analyses task for comment 4 and the parameter efficiency table for comment 6. However, the responses to two very important comments (1 & 2) on novelty and grounding of the proposed method are largely unsatisfactory. For novelty, applying/adopting an existing method to a new problem is difficult to accept as a novelty per ICLR standards. Likewise, it is important to know that how the proposed idea is bringing performance improvements claimed via some analyses and insights in the context of the problem solved and so it cannot be considered out-of-scope. In fact, the 1 & 2 are related in some sense, especially when the novelty aspect of the method can sometimes be justified by providing new and/or rigourous analyses.

---

> > > ### Author Response · Authors · 2023-11-22
> > > **Response to Reviewer ibUc**
> > >
> > > Thank you for your response and sharing your thoughts. We really appreciate them.
> > >
> > > - **Regarding "for novelty, applying/adopting an existing method to a new problem is difficult to accept as a novelty per ICLR standards."**\
> > > We respectfully disagree with your characterization that since we are adopting existing approaches to a new problem, it is trivial or not novel enough for ICLR. We believe our contribution is not that trivial and meets the ICLR standards of novelty. Our main contribution in this paper is to show that multimodal methods can be made robust to missing modalities by network adaptation. To the best of our knowledge, we are the first one to show that. Instead of proposing yet another method for network adaptation (or even worse, presenting an existing method with minor modification and giving it a new name), we performed a thorough and honest analysis with several existing adaptation methods, focused on the ones that we found effective, and reported the results. Furthermore, we presented results on a variety of tasks and datasets to demonstrate the robustness and versatility of our proposed approach. We believe our experiments and empirical analysis are detailed and rigorous in that sense.
> > >
> > >    Furthermore, ICLR has multiple papers in recent years that apply existing methods to solve new problems. A quick search gives us following notable papers:
> > >
> > >    **Paper [1]** accepted to ICLR 2021 combined supervised contrastive loss with cross entropy loss to fine tune pretrained models (i.e., applying existing supervised contrastive loss for model finetuning).
> > >
> > >    **Paper [2]** accepted to ICLR 2021 utilized entropy minimization (a well known technique) for test time adaptation.
> > >
> > >    **Paper [3]** accepted to ICLR 2022 showed that scaling strategy of transformer architecture (changing the number of layers, MLP dim etc) can affect the performance. The paper is an empirical study as claimed by the authors.
> > >
> > >    **Paper [4]** accepted in ICLR 2023 uses a model ensemble and show that it works well for knowledge transfer and better generalization.\
> > >    There are many more papers that use existing methods to solve new problems in ICLR.
> > > ######
> > > - **Regarding "how the proposed idea is bringing performance improvements claimed via some analyses and insights in the context of the problem solved and so it cannot be considered out-of-scope."**\
> > > We provided an explanation for this in our comment above. Our main motivation and insight for the proposed approach is to adapt the functional representation of the network according to the available modality combination. We have demonstrated the performance of our method empirically. We have provided detailed experiments with multiple datasets and tasks with ablation studies. We demonstrated that our method performs better than existing state-of-the-art robust methods. We included other experiments at your request because they are absolutely reasonable and certainly help the paper. What we meant is non-trivial and out-of-scope for one (largely empirical) paper is a rigorous mathematical analysis with say some achievable/expected error bounds on the functional representation. Such a theoretical analysis will inevitably have other issues such as unrealistic assumptions and mismatch with experiments. That is why we think it is beyond the scope of this paper.
> > > ****
> > > [1] Supervised Contrastive Learning for Pre-trained Language Model Fine-tuning - ICLR 2021 (https://iclr.cc/virtual/2021/poster/3275)\
> > >
> > > [2] Tent: Fully Test-Time Adaptation by Entropy Minimization - ICLR 2021 (https://iclr.cc/virtual/2021/poster/2874)\
> > >
> > > [3] Scale Efficiently: Insights from Pretraining and Finetuning Transformers - ICLR 2022 (https://openreview.net/forum?id=f2OYVDyfIB)\
> > >
> > > [4] Model ensemble instead of prompt fusion: a sample-specific knowledge transfer method for few-shot prompt tuning - ICLR 2023 (https://openreview.net/forum?id=p0yrSRbN5Bu)

---

### Official Review · Reviewer_wJeo · 2023-10-31

**Soundness:** 2 fair
**Presentation:** 3 good
**Contribution:** 3 good
**Rating:** 5
**Confidence:** 5

**Summary:**

This paper proposes a parameter-efficient adaptive method based on pre-trained multi-modal networks. This method recovers the missing modality by introducing a low-rank adaptive layer and a modulation scheme of intermediate features. Furthermore, this paper proves a simple linear operations can partially compensate for the performance degradation caused by missing modality. Finally, this paper conducts sufficient experiments on different datasets, showing the robustness and versatility of the proposed method.

**Strengths:**

1.This paper proposes an efficient approach to drive a multimodal model that adapts to different modality combinations, effectively addressing the performance degradation caused by missing modalities during testing.

2.The proposed method introduces only 0.7% of model parameters and achieves comparable performance to dedicated models on certain datasets.

3.Extensive experiments on multiple datasets demonstrate the superior performance of the proposed method compared to mainstream MML algorithms in handling the issue of missing modalities.

4.The writing of this paper is coherent, presenting clear contributions.

**Weaknesses:**

1.The method only uses zero to replace missing modalities, lacking other compensation methods such as duplicating available modal data or utilizing cross-modal generation techniques to fully showcase the impact of missing modalities.

2.The method's SSF strategy is an input-independent adaptation approach, it struggles to maintain reliable performance in scenarios with significant distribution differences between the training and testing sets.

3.Table 9 shows that there is still a gap compared to other adaptation method (BitFit), which makes the advantages of the proposed SSF strategy not obvious enough.

4.In Table 8 of the experimental section, the proposed method still exhibits a considerable performance gap compared to dedicated models. Can increasing the number of adaptation layer parameters further improve performance?

**Questions:**

See above weaknesses.

---

> ### Author Response · Authors · 2023-11-16
> **Response to Reviewer wJeo (Part 1/2)**
>
> Thank you for your detailed and insightful review and acknowledging the strengths of our paper. Below we respond to your other comments and questions.
>
> 1. **The method only uses zero to replace missing modalities, lacking other compensation methods such as duplicating available modal data or utilizing cross-modal generation techniques to fully showcase the impact of missing modalities.**\
>    We used zeros to replace missing modalities because that seems to be a common practice in recent papers \[1, 2]. We compared the performance of duplicating available modalities and replacing zeros for missing modalities in the table below (same settings as Table 1 in the paper). The results show that duplicating available modalities is inferior to the adapted model (and in some cases worse than replacing zeros for missing modalities). For MFNet and NYUDv2 datasets, we replaced the missing modality with a copy of the available modality. For the MCubeS dataset, we replaced the missing modalities with a copy of RGB modality. In all the combinations, the adapted model works much better than the modality duplication method.
>
>    |         |             |         |                                |                                    |           |
>    | :-----: | :---------: | :-----: | :----------------------------: | :--------------------------------: | :-------: |
>    | Dataset |    Input    | Missing | Pretrained (Zeros for missing) | Pretrained (Duplicate for missing) |  Adapted  |
>    |  MFNet  | RGB-Thermal |    -    |              60.10             |                  -                 |     -     |
>    |         |     RGB     | Thermal |              53.71             |              52.33             | **55.22** |
>    |         |   Thermal   |   RGB   |              35.48             |                44.43               | **50.89** |
>    |  NYUDv2 |  RGB-Depth  |    -    |              56.30             |                                    |           |
>    |         |     RGB     |  Depth  |              51.19             |              46.19             | **52.82** |
>    |         |    Depth    |   RGB   |              5.26              |                13.94               | **36.72** |
>    |  MCubeS |  RGB-A-D-N  |    -    |              51.54             |                  -                 |     -     |
>    |         |   RGB-A-D   |    N    |              49.06             |                49.93               | **51.11** |
>    |         |    RGB-A    |   D-N   |              48.81             |                49.23               | **50.62** |
>    |         |     RGB     |  A-D-N  |              42.32             |                48.96               | **50.43** |
>
>    Utilizing cross-modal generation is also a very good idea for missing modality compensation. Table 3 in our paper provides a comparison with one such method, TokenFusion, which dynamically detects uninformative tokens and substitutes these tokens with projected and aggregated tokens from available modalities. The CRM method in Table 2 uses self-distillation between the clean and masked modalities to learn complementary and non-local representations for better performance on missing modality scenarios. Our method performs better than both these methods in most of the scenarios.
> ######
> 2. **The method's SSF strategy is an input-independent adaptation approach; it struggles to maintain reliable performance in scenarios with significant distribution differences between the training and testing sets.**\
>    This is a good point and SSF may perform poorly in some cases, but in our experience SSF provides quite robust adaptation. According to \[3], parameters learnt by SSF are input-independent that makes it more suitable for representing the distribution of the different downstream dataset by modulating the intermediate feature. We could learn the modulation parameters by conditioning them on the input features, but that would require some additional modules (e.g. MLP and activation functions) at the expense of additional parameters.
> ****
> \[1] Mengmeng Ma, Jian Ren, Long Zhao, Davide Testuggine, Xi Peng; “Are Multimodal Transformers Robust to Missing Modality?”, Proceedings of the IEEE/CVF Conference on Computer Vision and Pattern Recognition (CVPR), 2022, pp. 18177-18186
>
> \[2] Sangmin Woo, Sumin Lee, Yeonju Park, Muhammad Adi Nugroho, Changick Kim;
> “Towards Good Practices for Missing Modality Robust Action Recognition.”, AAAI 2023: 2776-2784
>
> \[3] Lian, Dongze and Zhou, Daquan and Feng, Jiashi and Wang, Xinchao; “Scaling & Shifting Your Features: A New Baseline for Efficient Model Tuning”, Advances in Neural Information Processing Systems (NeurIPS), 2022

---

> > ### Author Response · Authors · 2023-11-16
> > **Response to Reviewer wJeo (Part 2/2)**
> >
> > 3. **Table 9 shows that there is still a gap compared to other adaptation methods (BitFit), which makes the advantages of the proposed SSF strategy not obvious enough.**\
> > The gap between BitFit and SSF is really small (0.1-0.2%). If we analyze all the results in Tables 4, 8, 9, and 10, the advantage of SSF would become obvious. BitFit provides better mIoU in two scenarios in Table 9, whereas SSF provides better mean accuracy and F1 scores. If we compare overall performance on all the three datasets (NYUDv2, MFNet and MCubeS), SSF works better than other methods for most of the scenarios as shown in Tables 4, 8, 9 and 10. For this reason, we preferred SSF as it can be applied to different datasets with diverse modalities.
> > ######
> > 4. **In Table 8 of the experimental section, the proposed method still exhibits a considerable performance gap compared to dedicated models. Can increasing the number of adaptation layer parameters further improve performance?**\
> > In principle, we can increase the number of adaptation modules (in attention and projection layers) to improve the performance in Table 8. In our experiments, we wanted to test the same adaptation procedure for different datasets for uniformity; therefore, we did not optimize the selection of adaptation modules for each experiment. As we can see in Table 9, 10, SSF performs better than the dedicated training approach for the other two datasets. Dedicated training trains the entire model with all the available input modalities, and that can be an upper bound in some cases.

---

> > > ### Author Response · Authors · 2023-11-22
> > >
> > > Dear Reviewer,\
> > > We hope you had a chance to read our response to your questions. We appreciate your feedback. Please let us know if you have any additional question that we can answer today. Thank you for your kind attention.

---

### Official Review · Reviewer_WPGU · 2023-11-01

**Soundness:** 2 fair
**Presentation:** 2 fair
**Contribution:** 2 fair
**Rating:** 3
**Confidence:** 4

**Summary:**

This paper proposes a method for multimodal learning with missing modalities. The writing is good. The experiments are suffcient. However, the math behind it is not clear, and the author does not give a reason why the data with missing modalities is not considered in the training phase.

**Strengths:**

The writing is good. The experiments are suffcient.

**Weaknesses:**

The math behind this paper is not clear, and the author does not give a reason why the data with missing modalities is not considered in the training phase.

**Questions:**

The datasets are not diverse enough. The authors only considered segmentation and sentiment analysis problems, different types of data should be considered. Figure 1 is not clear.

---

> ### Author Response · Authors · 2023-11-16
> **Response to Reviewer WPGU**
>
> Thank you for your review. We address your comments below.
>
> 1. **The math behind this paper is not clear.** \
> The main problem formulation is presented in Sec. 3. We believe we provided a concise mathematical description of our approach without unnecessary details. Equation 1 can be viewed as a model/network that is trained for all modalities $\mathcal{M}$ (consider it a baseline model). Equation 2 can be viewed as a model that is trained for a subset of modalities $\mathcal{S}$. We can in principle train one such model for every possible modality combination (dedicated training), but that would be prohibitive with a large number of possible modality combinations. Equation 3 represents our approach where we can learn a small set of parameters for different missing modality situations to adapt an already trained model. Mathematical formulations for different adaptation techniques are presented in Sec. 3.2. Please let us know if you have any other questions, and we will be happy to elaborate further.
> ######
> 2. **The author does not give a reason why the data with missing modalities is not considered in the training phase.** \
> It is unclear what you mean by this comment. We suspect you mean that we could use data with missing modalities at the training stage and either train a single robust network that performs well with all possible missing modalities or train one network for every specific missing modality combination. The latter approach is obviously infeasible as the number of networks required to train can become large. The former approach is something we considered and compared against (see results in Table 2 for VPFNet, SpiderMesh, CRM and Table 3 for TokenFusion). Our proposed network adaptation method clearly outperforms a single network trained to perform well with all possible missing modalities.\
> In our experiments, we wanted to separate the effects of robust training and network adaptation, but our proposed network adaptation scheme can be easily combined with a network pretrained in a robust manner. Furthermore, we compared our method with CRM in Table 2 that is trained in a robust manner and performs worse than our method on an average.
> ######
> 3. **The datasets are not diverse enough. The authors only considered segmentation and sentiment analysis problems, different types of data should be considered.** \
> We performed experiments for 3 different tasks (semantic segmentation, material segmentation and sentiment analysis). The tasks involve 5 different datasets and a total of 8 different modalities (RGB images, thermal images, depth maps, angle of linear polarization, degree of linear polarization, video, audio and text). For semantic segmentation we considered 2 different datasets: MFNet (RGB-thermal) and NYUDv2 (RGB-depth). For material segmentation we considered the MCubeS dataset that has four modalities: RGB images, thermal images, angle of linear polarization and degree of linear polarization. Multimodal sentiment analysis datasets (CMU-MOSI and CMU-MOSEI) have video, audio and text as modalities. We believe these datasets are diverse enough for a single paper. Please let us know if you have any specific dataset or task in mind. We will be happy to address that.
> ######
> 4. **Figure 1 is not clear.** \
> Figure 1 illustrates our overall approach to missing modality adaptation. Figure 1(a) shows that we can use a pretrained model, freeze all the weights, and insert adaptable layers after linear, convolutional, and norm (both batchnorm and layernorm) layers. Frozen and adaptable layers are illustrated with different colors for better understanding. Figure 1(b,c) shows two types of parameter-efficient adaptation schemes that can be used to modify the network output. We explained the figure in the first paragraph of Section 3.3 and explained the overall approach in Section 3.1 Equation 3. Please let us know which part is still not clear and we will explain or modify it as necessary.

---

> ### Author Response · Authors · 2023-11-22
>
> Dear Reviewer,\
> We hope you had a chance to read our response to your questions. We appreciate your feedback. Please let us know if you have any additional question that we can answer today. Thank you for your kind attention.

---

### Author Response · Authors · 2023-11-21

Dear reviewers,

We hope you had a chance to read our response. We tried to address all your concerns with explanations and additional experiments. We have also updated the paper pdf with additional experiments that compare modality duplication and zero replacement for missing modalities with our adaptation method in Table 12 (Section F in the supplementary material). We have also added comparison with existing methods for multimodal sentiment analysis in Table 13 (Section H in the supplementary material). Please let us know if you have any other questions or concerns.

Thank you again for your helpful comments and questions!

---

### Meta-Review · Area_Chair_wHtZ · 2023-12-09

**Metareview:**

The paper is concerns with learning multimodal models that are robust to missing modalities. The paper addresses this problem by low rank parameter formulation in certain layers that are adapted for missing modalities for downstream tasks.

The paper addresses an important problem, proposes a straightforward solution, and is well written.

However, all reviewers find important issues with the work. In particular, they are concerned with the diversity of used datasets, lack of comparisons to other approaches, and lack of sound insights of why the approach works for the proposed problem.

**Justification For Why Not Higher Score:**

The reviews are 1 x reject and 3 x borderline reject. Although they appreciate the clarity and quality of the manuscript, the reviewers aren't convinced by the choice of experiments, lack of comparisons, limited understanding of the workings of the algorithm. Hence, the paper is rejected from ICLR 2024.

**Justification For Why Not Lower Score:**

n/a

---

### Decision · Program_Chairs · 2024-01-16

Reject